# Highly selective electrocatalytic alkynol semi-hydrogenation for continuous production of alkenols

Jun Bu[1,2], Siyu Chang[2], Jinjin Li[2], Sanyin Yang[2], Wenxiu Ma[2], Zhenpeng Liu[1], Siying An[2], Yanan Wang[3], Zhen Li[1] & Jian Zhang [1,2] ✉

Alkynols semi-hydrogenation is a critical industrial process as the product, alkenols, have extensive applications in chemistry and life sciences. However, this class of reactions is plagued by the use of high-pressure hydrogen, Pd-based catalysts, and low efficiency of the contemporary thermocatalytic process. Here, we report an electrocatalytic approach for selectively hydrogenating alkynols to alkenols under ambient conditions. For representative 2-methyl-3-butene-2-ol, Cu nanoarrays derived electrochemically from CuO, achieve a high partial current density of 750 mA cm$^{-2}$ and specific selectivity of 97% at −0.88 V vs. reversible hydrogen electrode in alkaline solution. Even in a large two-electrode flow electrolyser, the Cu nanoarrays deliver a single-pass alkynol conversion of 93% with continuous production of 2-methyl-3-butene-2-ol at a rate of ~169 g g$_{Cu}$$^{-1}$ h$^{-1}$. Theoretical and in situ electrochemical infrared investigations reveal that the semi-hydrogenation performance is enhanced by exothermic alkynol adsorption and alkenol desorption on the Cu surfaces. Furthermore, this electrocatalytic semi-hydrogenation strategy is shown to be applicable to a variety of alkynol substrates.

The manufacture of chemicals accounts for ~8% of global energy consumption[1]. Alkenols, e.g., 2-methyl-3-buten-2-ol (MBE) and 2-butene-1,4-diol (BED), are essential intermediates and building blocks for numerous vital fine chemicals, including vitamins (A, B6, and E), pharmaceuticals, agrochemicals, fragrances, flavours, etc[2]. In recent decades, alkenols have been produced through thermocatalytic semi-hydrogenation reactions with corresponding alkynols (Fig. 1a). Accordingly, extensive efforts including elemental doping[3,4], crystal structure engineering[5], alloys or intermetallics[6–9] and support effects[10–12], have been dedicated to improving the activity and selectivity of the thermocatalytic semi-hydrogenation of alkynols. Unfortunately, the thermocatalytic semi-hydrogenation of alkynols still demonstrate several major disadvantages: 1) pressurized hydrogen gas (1–30 bars) must be used as the hydrogen source, which leads to

potential safety issues and overhydrogenation of alkynols to alkanols; 2) precious metal Pd-based catalysts, such as Lindlar catalysts, are imperative for the process; 3) elevated reaction temperatures (< 160 °C) must be provided for boosting the alkynol conversion; 4) the undesirable hydrogenolysis or hydrodeoxygenation of alkynols generate by-products that are difficult to separate; and 5) toxic Pb additives in the catalysts causes severe environmental problems and contaminates the alkenols with heavy metals. Therefore, a selective alkynol semi-hydrogenation strategy that is efficient and environmentally friendly is urgently needed for the cost-effective production of alkenols.

Recently, owing to their mild conditions, ever-growing attention has been placed on the utilization of renewable electricity to manufacture liquid fuels[13,14] (alcohols, etc.) and commodity chemicals[15–18]

[1]State Key Laboratory of Solidification Processing and School of Materials Science and Engineering, Northwestern Polytechnical University, Xi'an, Shaanxi 710072, PR China. [2]Key Laboratory of Special Functional and Smart Polymer Materials of Ministry of Industry and Information Technology and School of Chemistry and Chemical Engineering, Northwestern Polytechnical University, Xi'an, Shaanxi 710129, PR China. [3]Hualu Engineering and Technology Co., Ltd, Xi'an, Shaanxi 710065, PR China. ✉e-mail: zhangjian@nwpu.edu.cn

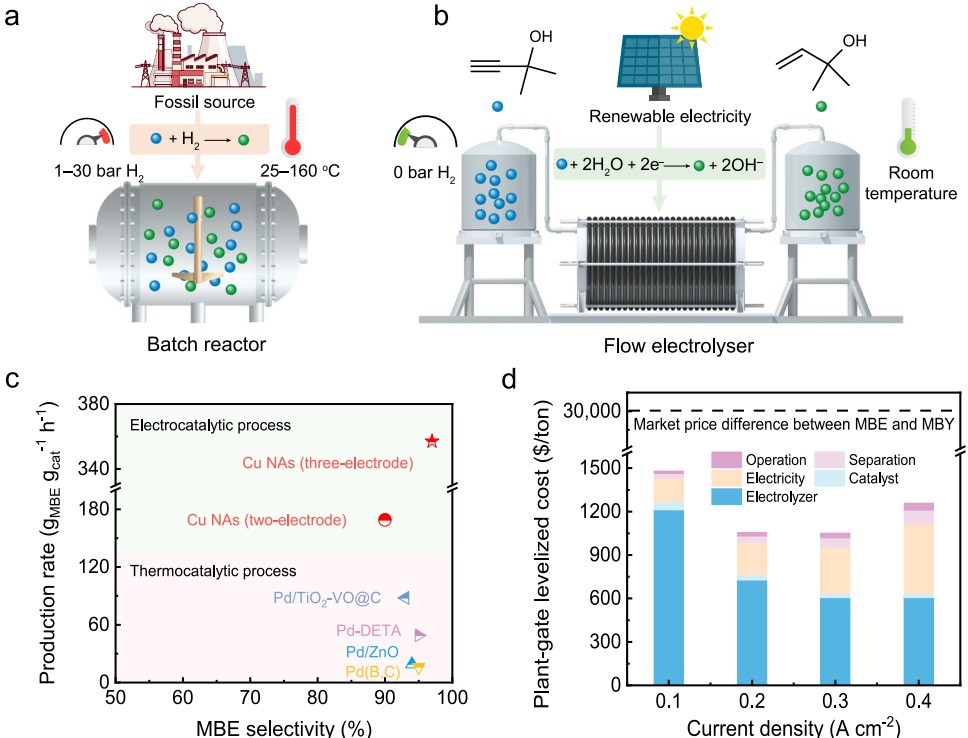

**Fig. 1 | Electrocatalytic semi-hydrogenation of MBY by using renewable energy.** Schematic illustrations: (**a**) the industrial thermocatalytic process and (**b**) the proposed electrocatalytic process. **c** Comparison of the thermocatalytic and electrocatalytic processes. **d** Breakdown of the plant-gate levelized cost per ton for the MBE produced over Cu nanoarrays (Cu NAs) at different current densities, as calculated from the technoeconomic analysis. The TEA calculation details are provided in Supplementary Note 1.

(ethylene, ethylene oxide, hydrogen peroxide, etc.). In comparison with the energy-intensive thermocatalytic process, the selective electrochemical semi-hydrogenation of alkynols to alkenols, in which water molecules serve as the hydrogen source, is an appealing alternative. Previously, to understand the mechanism of hydrogenation by heterogeneous thermocatalysts, the adsorption and hydrogenation behaviour of 2-propyn-1-ol and 2-methyl-3-butyne-2-ol (MBY) on a single-crystal Pt electrode were qualitatively investigated in acidic solutions by using cyclic voltammetry (CV) and in situ Raman spectroscopy[19]. However, the electrocatalytic semi-hydrogenation of alkynols still remains unexplored due to the lack of high-efficiency electrocatalysts and strong competition of hydrogen evolution (HER) and overhydrogenation reactions.

In this work, we report an electrocatalytic approach to perform the semi-hydrogenation of alkynols for the continuous production of alkenols with high activity and selectivity at ambient temperature and pressure (Fig. 1b and Supplementary Fig. 1. As a result, with Cu NAs (electrochemically derived from CuO) on Cu foam as cathodic electrocatalysts, the Faradaic efficiency of MBE (FE$_{MBE}$) reaches 95% at −0.15 V vs. RHE in 1 M KOH aqueous solution containing 0.5 M MBY. Even at an extremely high current density of 1.3 A cm$^{-2}$ (−0.88 V vs. RHE), the partial current density ($j_{MBE}$) and MBE selectivity remarkably reach 750 mA cm$^{-2}$ and ~97%, respectively, corresponding to a MBE production rate of 357 g g$_{Cu}^{-1}$ h$^{-1}$, which substantially exceeds the reported values for state-of-the-art thermocatalysts[4,7,11,20] (Fig. 1c). Over 20 runs (in which 840 mg MBY is circularly fed in 20 mL of 1 M KOH aqueous solution for each run), the MBY conversion and MBE selectivity are stably maintained at > 90%. Importantly, in a large-area two-electrode flow electrolyser (25 cm$^2$) delivered with a 0.5 M MBY-containing alkaline electrolyte at a flow rate of 134 mL h$^{-1}$, an continuous MBE production rate of 169 g g$_{Cu}^{-1}$ h$^{-1}$ is achieved. In addition, this electrocatalytic semi-hydrogenation strategy is universally applicable to the selective semi-hydrogenation of other alkynols,

including primary, secondary, and tertiary alcohols, as well as diols. For 1 tonne of MBE, the technoeconomic analysis (TEA) calculations indicate that the plant-gate levelized cost of electrocatalytic MBY semi-hydrogenation (<1500 $/ton) is much lower than the large market price difference between MBE and MBY (~30,000 $/ton), demonstrating the great potential of this semi-hydrogenation process for replacing conventional thermocatalytic processes (Fig. 1d).

## Results and discussion

### Structural characterization and electrocatalytic performance of the Cu NAs

According to our previous work on electrocatalytic acetylene reduction, carbon-carbon triple bonds and carbon-carbon double bonds present a strong σ-π configuration and a weak π configuration on Cu surfaces, respectively, which endow Cu catalysts with excellent intrinsic properties for adsorbing acetylene and desorbing ethylene[15]. Earth-abundant Cu electrocatalysts are thus postulated for selectively hydrogenating alkynols to alkenols using water as hydrogen source. Accordingly, Cu NAs, which feature abundant active sites, were fabricated on commercial Cu foam (Supplementary Fig. 2). Cu(OH)$_2$ NAs were synthesized beforehand on Cu foam through chemical oxidation[21]. After heat treating the Cu(OH)$_2$ NAs at 150 °C in air, CuO NAs formed on the Cu foam. Eventually, the CuO NAs were electrochemically reduced in situ to Cu NAs in 1 M KOH aqueous electrolyte. The X-ray diffraction (XRD), X-ray photoelectron spectroscopy (XPS), scanning electron microscopy (SEM) and transmission electron microscopy (TEM) characterization results confirmed that Cu NAs formed on the Cu foam (Fig. 2a, b and Supplementary Figs. 3–5). The as-fabricated Cu NAs had diameters of ~100–200 nm and were composed of aggregated nanoparticles. The high-resolution TEM results revealed that the Cu NAs exposed an abundance of (111) facets with a lattice distance of ~2.08 Å (Supplementary Fig. 6).

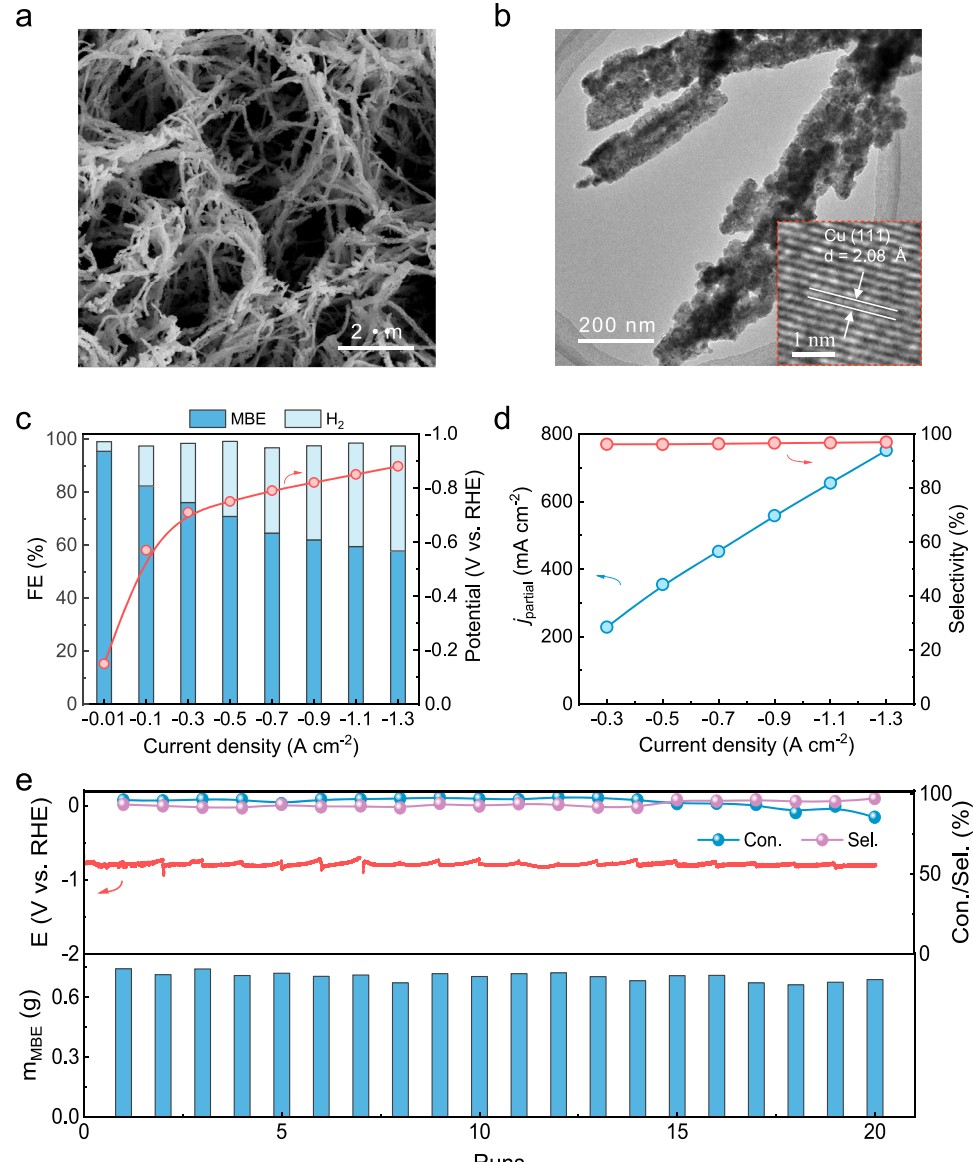

**Fig. 2 | Electrocatalytic semi-hydrogenation performance of the Cu NAs. a** SEM images of Cu NAs. **b** TEM and HRTEM (inset) images of Cu NAs. **c** Faradaic efficiency distributions of the products and corresponding applied potentials at different current densities in the 1 M KOH solution containing 0.5 M MBY. **d** Partial current density and selectivity of MBE at different current densities. **e** Stability evaluation of the Cu NAs over 20 runs at a current density of 1.3 A cm⁻², in which 20 mL of the 1 M KOH aqueous solution containing 0.5 M MBY was circularly fed for each run.

Next, the electrocatalytic hydrogenation performance of the Cu NAs was experimentally evaluated in a three-electrode flow cell. Ni foam and Hg/HgO were utilized as the counter and reference electrodes, respectively. An anion exchange membrane (AEM) separated the anodic and cathodic chambers. As a typical alkynol, MBY was employed as the probe to investigate the electrocatalytic performance of the Cu NAs. In a 1 M KOH aqueous solution without MBY, the cathodic current density of the Cu NAs was ~0.8 A cm⁻² at −0.7 V vs. RHE, which was completely attributed to the HER process. Remarkably, after the addition of 0.5 M MBY to the 1 M KOH aqueous solution, the current density of the Cu NAs dramatically decreased to ~0.3 A cm⁻² at −0.7 V; this result undoubtedly suggested that MBY effectively suppressed the HER process (Supplementary Fig. 7). The products of electrocatalytic MBY hydrogenation were qualitatively confirmed to be MBE through the use of gas chromatography-mass spectrometry (GC-MS) (Supplementary Fig. 8). Chronopotentiometry experiments were conducted to evaluate the electrocatalytic performance (Supplementary Fig. 9). As a result, the FE_MBE of Cu NAs was

~95% at 10 mA cm⁻², which was far larger than ~4% for hydrogen FE (Fig. 2c). These results unambiguously demonstrated that the electrocatalytic activity of the semi-hydrogenation of MBY on the Cu NAs was intrinsically superior to that of the HER process. In addition, the Cu NAs exhibited a j_MBE of 228 mA cm⁻² and a FE_MBE of 76% at 0.3 A cm⁻², which were considerably higher than 131 mA cm⁻² and 44% for the electrochemically deposited Pd nanoparticles on the Cu foam (Supplementary Figs. 10 and 11). With increasing current densities, the FE_MBE of the Cu NAs gradually decreased from ~80% at 0.1 A cm⁻² to ~60% at 1.1 A cm⁻² as a result of the increasing competition of the HER. Nevertheless, the j_MBE markedly increased from ~230 to ~750 mA cm⁻² when the applied current density was improved from 0.3 to 1.3 A cm⁻², and these density values were notably larger than the industrially required current density of 200 mA cm⁻² (Fig. 2d)²². Even at a current density of 1.3 A cm⁻², the FE_MBE and MBE selectivity still reached as high as 58% and 97%, respectively, indicating that the overhydrogenation of MBY to 2-methyl-3-butan-2-ol (MBA) was effectively suppressed. Even at full MBY conversion (>99%), the MBE selectivity was 91%

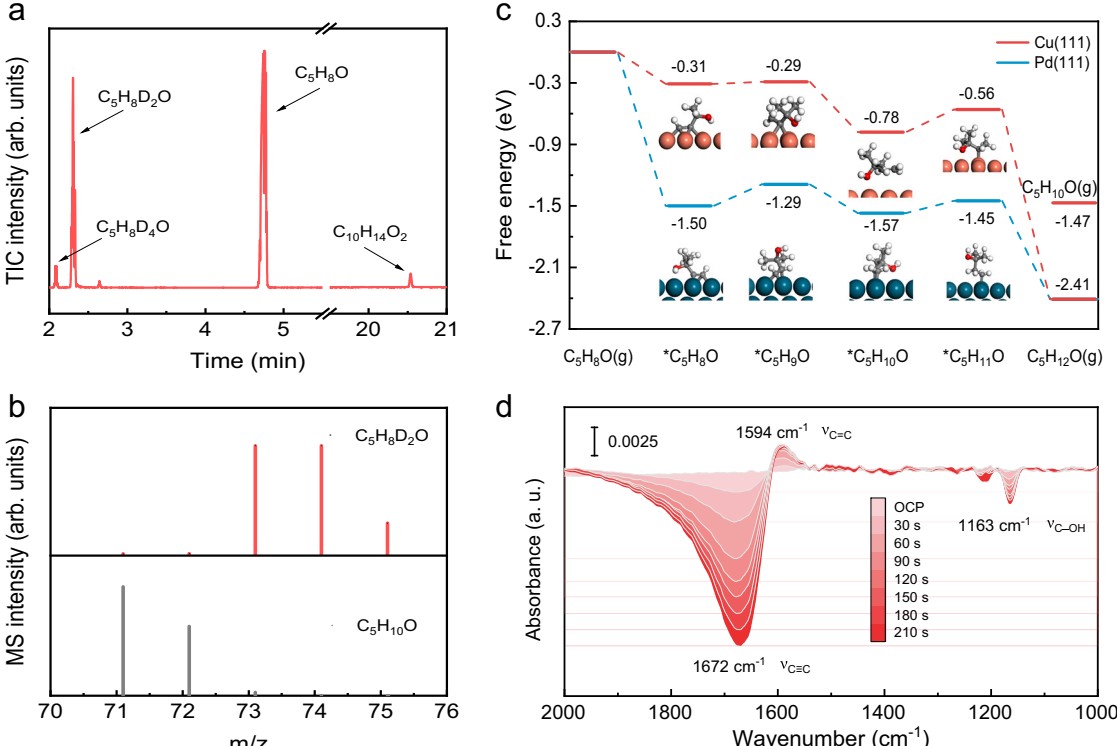

**Fig. 3 | Theoretical and experimental investigations on the reaction mechanism. a** Total ion chromatogram (TIC) of the liquid products in the D isotope labelling experiment. **b** Mass fragmentation patterns (red bars) were acquired at a 2.3 min retention time of the TIC in Fig. 3a. For comparison, a reference fragmentation pattern of $C_5H_{10}O$ (MBY) (grey bars) is displayed. **c** Free energy diagrams of the MBY hydrogenation process on Cu(111) and Pd(111) surfaces. The inserts are the optimal structures of *$C_5H_8O$, *$C_5H_9O$, *$C_5H_{10}O$ and *$C_5H_{11}O$ on Cu(111) (top) and Pd(111) (bottom). **d** In situ electrochemical ATR-FTIR spectra obtained during the electrocatalytic semi-hydrogenation of MBY on the Cu NAs.

(Supplementary Fig. 12). The MBE production rate of the Cu NAs at 1.3 A cm$^{-2}$ reached 357 g g$_{cat}^{-1}$ h$^{-1}$ (Supplementary Fig. 13), which was much higher than the values for state-of-the-art thermocatalysts, e.g., 16 g g$_{cat}^{-1}$ h$^{-1}$ for Pd(B, C)[4], 19 g g$_{cat}^{-1}$ h$^{-1}$ for Pd/ZnO[7], 49 g g$_{cat}^{-1}$ h$^{-1}$ for Pd-DETA[20] and 88 g g$_{cat}^{-1}$ h$^{-1}$ for Pd/TiO$_2$-VO@C[11]. Regarding industrial water-splitting electrolysers, improving the reaction temperature is necessary to enhance catalytic activity[23]. As expected, the Cu NAs prominently delivered a particularly high $j_{MBE}$ of -1.3 A cm$^{-2}$ at 60 °C (Supplementary Fig. 14), while the corresponding FE$_{MBE}$ and MBE selectivity remained at 67% and 96%, respectively. In addition, electrocatalytic MBY hydrogenation performance of different Cu-based catalysts was evaluated. Obviously, the MBE selectivity and the electrochemical active surface area (ECSA) normalized $j_{MBE}$ of different Cu-based electrocatalysts were similar. This result reveals that the electrocatalytic MBY hydrogenation performance is largely insensitive to the Cu structure (Supplementary Fig. 15).

Electrocatalytic stability is another important criterion for assessing electrocatalyst performance. Thus, a recycling experiment of the electrocatalytic semi-hydrogenation of MBY on the Cu NAs was performed at a high current density of 1.3 A cm$^{-2}$ in a three-electrode flow cell. As shown in Fig. 2e, due to the excellent electrocatalytic performance of the Cu NAs, 840 mg of MBY (10 mmol) in 20 mL of 1 M KOH solution was efficiently hydrogenated into 750 mg of MBE after the first electrocatalytic run (1 h for one run), corresponding to a MBY conversion of 96% and a MBE selectivity of 93%. With an increasing number of recycling runs (up to 20), the MBY conversion of the Cu NAs slightly decreased from 96% to 91%. However, the MBE selectivity remained at -95%; additionally, the changes in applied potential were negligible. For each run, even if the MBY concentration gradually decreased over the reaction time, the average FE$_{MBE}$ and $j_{MBE}$ still reached 33% and 420 mA cm$^{-2}$, respectively (Supplementary Fig. 16).

The SEM, TEM, XRD and XPS investigations revealed that no obvious morphological and structural variations were observed for the Cu NAs after the long-term electrocatalytic stability test (Supplementary Figs. 17–20).

## Reaction mechanism of electrocatalytic alkynol semi-hydrogenation

Isotope labelling experiments were first employed to trace the hydrogen source for the electrocatalytic semi-hydrogenation of alkynols (Fig. 3a, b)[24]. D$_2$O was used instead of H$_2$O. After the electrocatalytic hydrogenation of alkynols was performed on the Cu NAs in 1 M KOH D$_2$O solution, GC−MS analyses were conducted; the results confirmed the characteristic mass fragmentation patterns of $C_5H_8D_2O$, which shifted to a higher m/z compared with that of the reference fragmentation patterns of MBE. Thus, the hydrogen source for electrochemical hydrogenation of MBY is definitely from abundant water molecules.

The competitive mechanism between MBY hydrogenation and the HER was further studied by changing the MBY concentrations in the 1 M KOH aqueous solution. A MBY concentration of > 1.5 M was not evaluated because the anion exchange membrane was susceptible to crossover with a high concentration of organic molecules[25]. Notably, as the MBY concentration was increased from 0.1 to 1.5 M, the current densities of the Cu NAs drastically decreased from 0.88 to 0.18 A cm$^{-2}$ at −0.7 V (Supplementary Fig. 21a). Theoretical simulations showed that the free energy of MBY adsorption on the Cu(111) surface was −3.20 eV, which was much lower than the −0.33 eV for water adsorption (Supplementary Fig. 22). Thus, in comparison with H$_2$O, the MBY molecules preferentially bonded to the Cu NAs. The increasing MBY concentration expanded the MBY coverage on the Cu surfaces, which inherently hindered H$_2$O adsorption. At high MBY concentrations, HER

**Table 1 | Universality of the electrocatalytic semi-hydrogenation of alkynols on Cu NAs**

Alkynols were electrochemically hydrogenated at a current density of 1.3 A cm$^{-2}$ and a time of 15 minutes, except for **1c** and **1h**, which were hydrogenated for 30 and 60 minutes, respectively. Reaction conditions: 20 mL of 1 M KOH electrolyte; 2 mmol of alkynols; 1.0 cm$^2$ Cu NAs; room temperature.

kinetics were more strongly suppressed; thus, FE$_{MBE}$ correspondingly increased from 20% for 0.1 M MBY to 64% for 1 M MBY at 1.3 A cm$^{-2}$ (Supplementary Fig. 21b). Moreover, $j_{MBE}$ eventually reached 833 mA cm$^{-2}$ in 1 M MBY (Supplementary Fig. 21c). Nevertheless, MBE selectivity was independent of the MBY concentrations and always exceeded 90% (Supplementary Fig. 21d), proving the excellent suppression of overhydrogenation. The near exponential augment of $j_{MBE}$ was dependent on the applied potentials in 0.5 M MBY, revealing the minimal impact of the mass transport of MBY on the electrocatalytic activity (Supplementary Fig. 23)[26]. In principle, the pH plays a key role in the HER and hydrogenation kinetics[27]. Thus, various aqueous solutions with different pH values were investigated. In contrast to the above results, the Cu NAs exhibited noticeably decreased current densities and large potentials in 1 M KHCO$_3$ solution (pH = 8.5), which might be attributed to the competitive adsorption of HCO$_3^-$ on the Cu surfaces[28]. When the concentration of the alkaline electrolyte was gradually increased from 0.5 to 3 M, the $j_{MBE}$ plot shifted toward high potentials (Supplementary Figs. 24 and 25). Furthermore, different amount of KCl was added into 1 M KOH electrolyte. As shown in (Supplementary Fig. 26a), the pH value of the electrolytes was well kept at -13.7, but the electrical conductivity increased from 180 mS cm$^{-1}$ for 1 M KOH to 225 mS cm$^{-1}$ for 1 M KOH + 0.5 M KCl and 270 mS cm$^{-1}$ for 1 M KOH + 1 M KCl. As revealed in Supplementary Fig. 26b, c, the variations of MBY hydrogenation performance were negligible. These results confirmed that solution resistance had negligible influence on MBY hydrogenation performance. Therefore, the improvement of $j_{MBE}$ at rising pH values is possibly due to the accelerated water dissociation[29]. Cyclic voltammograms (CV) were performed for investigating the effects of the adsorbed H* on MBY

hydrogenation in 1 M KOH solution. As revealed in Supplementary Fig. 27a, the CV curve in pure 1 M KOH solution showed an anodic peak at about 0.23 V vs. RHE, which was attributed to H* desorption. By contrast, in 1 M KOH solution containing MBY, the intensity of H* desorption peaks remarkably decreased along with increased MBY concentration, indicating the consumption of surface H* species during MBY hydrogenation. Notably, the H* desorption peak well recovered in fresh KOH electrolyte (Supplementary Fig. 27b). The Tafel slopes of Cu NAs were 118 mV dec$^{-1}$ for HER and 114 mV dec$^{-1}$ for MBY hydrogenation (Supplementary Fig. 27c), suggesting that the Volmer step was the rate-determining step. Thereby, for MBY hydrogenation on Cu NAs, active H* was first generated via the Volmer step. To further verify the MBY hydrogenation mechanism on Cu NAs, the reaction rates were evaluated at high MBY concentrations (0.5 M to 1.5 M) (Supplementary Fig. 27d). A negative reaction order (−0.46) for MBY was achieved, meaning that the reaction rate declined along with increased the MBY concentration. This was due to the competitive adsorption of MBY and H$_2$O molecules on Cu NAs.

Theoretical simulations and in situ electrochemical attenuated total reflectance-Fourier transform infrared (ATR-FTIR) spectroscopy were further conducted to thoroughly examine the underlying reaction mechanism of the electrocatalytic semi-hydrogenation of MBY. Figure 3c shows the free-energy diagrams of MBY hydrogenation on the (111) surfaces of Cu and Pd catalysts. MBY adsorption on Cu and Pd surfaces was exothermic. The first hydrogenation step of converting the adsorbed MBY to *C$_5$H$_9$O on Cu was energetically uphill with a minor △Gr of only 0.02 eV; this result was substantially lower than the 0.21 eV for Pd. The subsequent exothermic hydrogenation process with *C$_5$H$_9$O further promoted the formation of MBE* on Cu. Notably,

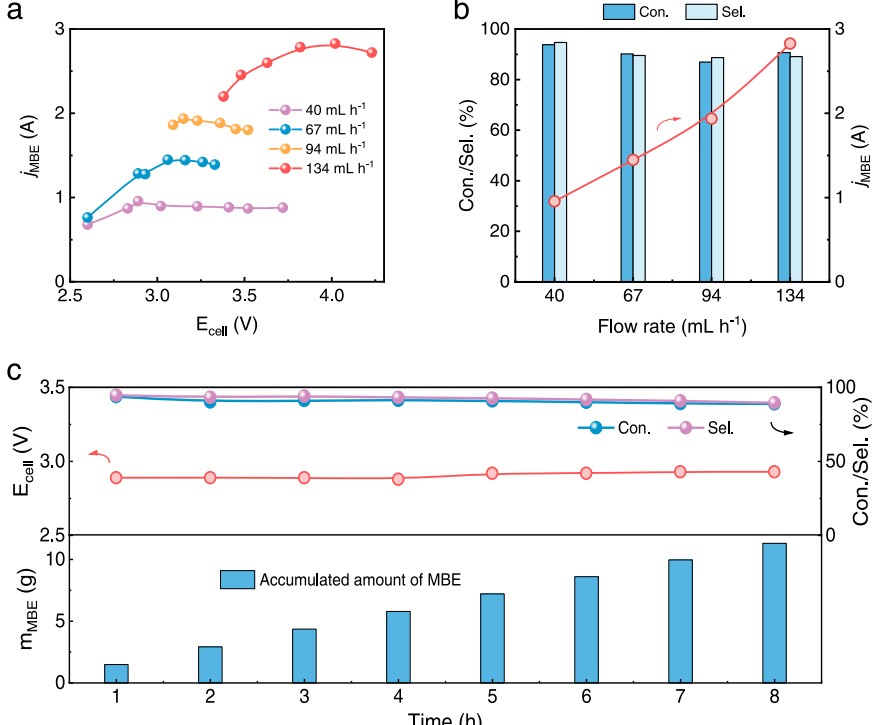

**Fig. 4 | Electrocatalytic semi-hydrogenation performance of MBY on the Cu NAs in a 5 × 5 cm² flow electrolyser. a** Partial current density of MBE at various cell voltages and electrolyte flow rates. **b** Maximum partial current density of MBE, corresponding MBY single-pass conversion and MBE selectivity at different electrolyte flow rates. **c** Continuous 8 h production of MBE at a cell current of 1.8 A by feeding 1 M KOH aqueous solution containing 0.5 M MBY with a flow rate of 40 mL h⁻¹.

the overhydrogenation of MBE* to *C₅H₁₁O was endothermic with a large △Gr of 0.22 eV. In contrast, the desorption of MBE from the Cu surfaces was energetically downhill (△Gr = −0.69 eV), which eventually facilitated rapid MBE desorption rather than its over-hydrogenation. Afterwards, in situ electrochemical ATR-FTIR spectra were recorded at 20 mA cm⁻² to investigate the reaction process (Fig. 3d)[30]. Without Cu electrocatalysts, the characteristic peaks of the carbon-carbon triple bonds and C−OH bonds in MBY in 1 M KOH solution were located at 2118 cm⁻¹ and 1163 cm⁻¹, respectively (Supplementary Fig. 28)[31]. Markedly, at an open circuit potential (OCP), the peak of the carbon-carbon triple bonds on Cu drastically shifted to 1672 cm⁻¹, while the signal of the C−OH bonds remained at 1163 cm⁻¹. These results revealed that the MBY molecules preferred to adsorb onto the Cu surfaces via a σ-π-configuration between the carbon-carbon triple bonds and surface Cu atoms, which agreed well with the theoretical model[32]. After switching to the electrocatalytic process, a distinct signal of the carbon-carbon double bonds in MBE appeared at 1594 cm⁻¹, which was different than the 1649 cm⁻¹ observed for free MBE in alkaline solution[32,33]. This slight red-shift of the carbon-carbon double bonds in MBE indicated the weak adsorption of MBE on the Cu surfaces, ultimately accelerating its rapid desorption from the Cu catalysts[34]. With increasing reaction time, the peak intensities of the carbon-carbon triple bonds decreased, while the peak intensities of the carbon-carbon double bonds increased (Supplementary Fig. 29). These results confirmed the occurrence of selective electrocatalytic semi-hydrogenation of MBY to MBE.

**Universality and practical implementation of electrocatalytic alkynol semi-hydrogenation**
To estimate the feasibility of the electrocatalytic semi-hydrogenation approach for other key alkynols, the electrocatalytic performance of the Cu NAs was measured in a three-electrode flow cell. A 1 M KOH aqueous solution containing different alkynols, including primary,

secondary and tertiary alcohols as well as diols, was utilized as the electrolyte. As described in Table 1, the Cu NAs universally exhibited excellent electrocatalytic conversion (≥ 80%) and alkenol selectivity (≥ 81%) for all alkynols. ¹H NMR spectra of the as-synthesized alkenols are shown in Supplementary Fig. 30. In particular, owing to notable steric hindrance during their adsorption on Cu surfaces, the electro-catalytic conversion of alkynols with nonterminal acetylenic bonds was generally lower than that for terminal alkynes. Even for 2,5-dimethyl-3-hexyne-2,5-diol, the Cu NAs still achieved a conversion of 80% and a very high semi-hydrogenation selectivity of 93%.

Regarding industrial implementation, upgrading from batch to continuous operation is of vital significance to reduce the number of processing steps, thereby enhancing production efficiency and alle-viating environmental impact. As the key parameter of atom economy, the E-factor can be decreased from 23 → 3 kg waste per kg product[8], and to achieve that, we customized a two-electrode flow electrolyser with a large electrode area of 25 cm² by integrating NiFe layered double hydroxides on Ni foam as the anode. This setup provided a substantial increase in residence time for the electrocatalytic semi-hydrogenation of alkynols. A 1 M KOH electrolyte containing 0.5 M MBY was con-tinuously delivered to the cathodic Cu NAs at different flow rates. The variations of polarization curves at different flow rates were negligible (Supplementary Fig. 31). With increasing cell voltages, MBY conversion gradually increased, while MBE selectivity and FE_MBE correspondingly decreased owing to increasing competition of the HER and over-hydrogenation reactions (Supplementary Fig. 32). Remarkably, with the single-pass electrocatalytic semi-hydrogenation process at a $j_{MBE}$ of 2.83 A and a flow rate of 134 mL h⁻¹, the Cu NAs exhibited a MBY con-version of ~90% and a MBE selectivity of ~90% (Fig. 4a, b). Additionally, 0.405 M MBE in a 1 M KOH solution was continuously generated at a rate of 169 g g_Cu⁻¹ h⁻¹, which was much higher than the rates of pre-viously reported thermocatalysts[4,7,11,20] (Supplementary Table 1). Dur-ing a 8 h stability operation at 1.8 A, the Cu NAs held a single-pass MBY

conversion of ~91% and a MBE selectivity of ~93% (Fig. 4c). The mass fraction of purified MBE product reached as high as 98% (Supplementary Fig. 33). The continuous production of MBE was also investigated at 3.0 A for 60 h with a single-pass MBY conversion of ~83% and a MBE selectivity of ~92%, eventually produced about 130 g MBE (Supplementary Fig. 34).

We have demonstrated an electrocatalytic semi-hydrogenation approach for selectively reducing alkynols to alkenols. This approach utilizes water as a hydrogen source and is performed at room temperature and ordinary pressure. The excellent alkynol adsorption and alkenol desorption properties of the Cu NAs are responsible for their high activity and selectivity that are superior to state-of-the-art thermocatalysts. In a large two-electrode flow electrolyser, the Cu NAs deliver a continuous alkynol production rate of up to ~169 g $g_{Cu}^{-1}$ h$^{-1}$, further showing its promise as a replacement for the relatively energy-intensive, high-cost, low-efficiency, and hazardous thermocatalytic alkynol hydrogenation process.

## Methods

### Materials

Commercial copper foam with a thickness of 1.6 mm was purchased from Changsha Lyrun New Materials Co. Ltd. Copper nitrate ($Cu(NO_3)_2 \cdot 3H_2O$, 98%), HCl solution (37 wt%), acetone, and ethanol were purchased from Guangdong Guanghua Sci-Tech Co., Ltd. Potassium hydroxide (KOH, 99.99%), sodium hydroxide (NaOH, 98%), palladium chloride ($PdCl_2$, 99%) and ammonium thiosulfate ($(NH_4)_2S_2O_8$, 98%) were purchased from Shanghai Aladdin Bio-Chem Technology Co., Ltd. The anion exchange membrane (Fumasep FAB-PK-130) was purchased from Fuel Cell Store. All chemical reagents were used as received without further purification. The ultrapure water (> 18.25 MΩ cm) was utilized for the experiments.

### Preparation of Cu nanoarrays

Typically, commercial copper foam (thickness of 1.6 mm) was cut into the pieces of 1.0 × 3.0 cm$^2$, which were then consecutively cleaned by using 1 M HCl solution, acetone, and ethanol, rinsed with distilled water, and eventually blow-dried with nitrogen. In a typical procedure, 2.40 g NaOH (60 mmol) and 1.83 g ($NH_4)_2S_2O_8$ (8 mmol) were first dissolved into 40 mL distilled water and then stirred to form a clear solution. Subsequently, the copper foam was immersed into the above mixed solution for 20 min at room temperature. After that, the as-formed $Cu(OH)_2$ nanoarrays on the copper foam was rinsed with de-ionized water and absolute ethanol, and dried with nitrogen. CuO nanoarrays on the copper foam were then fabricated by annealing the $Cu(OH)_2$ nanoarrays at 150 °C for 1 h in air. Eventually, the CuO nanoarrays were in-situ electrochemically reduced into Cu nanoarrays in 1 M KOH aqueous electrolyte at −1.0 A cm$^{-2}$ for 10 min, using Hg/HgO electrode and nickel foam as the reference and counter electrodes, respectively. The loading weight of Cu nanoarrays on the copper foam was estimated to be 3.30 mg cm$^{-2}$ according to the consuming charge for the reduction of CuO to Cu[21]. The Cu nanoarrays used in the two-electrode flow electrolyser with a large electrode area of 25 cm$^2$ (5 × 5 cm$^2$) was synthesized in the same way, except for 7.20 g NaOH (0.18 mol) and 5.47 g ($NH_4)_2S_2O_8$ (0.024 mol) were first dissolved into 200 mL distilled water to synthesize the catalyst. Eventually, the Cu nanoarrays were in-situ electrochemically reduced in 1 M KOH aqueous electrolyte at 10.0 A for 10 min in the electrolyser (25 cm$^2$) before the full cell testing. In this case, the mass loading of Cu nanoarrays on the copper foam was about 1.05 mg cm$^{-2}$.

### Preparation of the electrodeposited Pd nanoparticles

The Pd nanoparticles were electrodeposited onto copper foam using a one-compartment electrochemical cell[35]. A piece of copper foam with an exposed geometric surface area of 1.0 × 3.0 cm$^2$ served as a working electrode with reference to the Ag/AgCl reference electrode and Pt mesh counter electrode. The compartment was filled with 50 mL 1 M HCl electrolyte containing 15.9 mM $PdCl_2$. A current density of −200 mA cm$^{-2}$ was applied to the copper foam for 50 s to reduce Pd ions in solution. The Pd catalyst on the cooper foam was then rinsed with de-ionized water and absolute ethanol, and finally flow-dried with nitrogen. The loading weight of Pd nanoparticles on the copper foam was estimated to be about 3.51 mg cm$^{-2}$.

### Electrocatalytic evolution

We designed two electrolyser cells with the window areas of 1 cm$^2$ (2 × 0.5 cm$^2$) and 25 cm$^2$ (5 × 5 cm$^2$). Both of them were built in-house and consisted of gaskets, anode and cathode flow-field plates. The anode and cathode plates were separated by the anion exchange membrane (AEM, Fumasep FAB-PK-130) and individually delivered with the electrolyte. The catholyte consisted of 0.5 M MBY in 1 M KOH aqueous solution while the anolyte was 1 M KOH aqueous solution. Electrochemical measurements were performed using electrochemical workstations (Corrtest CS150M). The Cu nanoarrays on copper foam were used as the working electrode. Hg/HgO electrode and nickel foam were the reference and counter electrodes, respectively. For electrochemical tests, all the potentials were provided without iR-compensation and were converted into the RHE according to: E (versus RHE) = E (versus Hg/HgO) + 0.098 V + 0.059 V × pH.

Chronopotentiometry experiments were conducted at −0.01, −0.1, −0.3, −0.5, −0.7, −0.9, −1.1, −1.3 A cm$^{-2}$, respectively, to measure the Faradaic efficiency and selectivity of Cu nanoarrays. For each data, the electrolysis was carried out for 600 s to collect the products before injected into the gas chromatography (GC). The gaseous products were analyzed by gas chromatography (Techcomp GC7900), equipped with a thermal conductivity detector (TCD) and a flame ionization detector (FID). The liquid products were extracted with ethyl acetate and then quantified by gas chromatography (Fuli GC9790Plus) equipped with a flame ionization detector (FID). The faradaic efficiency (FE) of products was computed according to the following equation:

$$FE(\%) = \frac{nmF}{It} \times 100 \qquad (1)$$

where $n$ = number of transferred electrons
$m$ = amount of substance
$F$ = Faraday's constant
$I$ = total current
$t$ = electrolysis time

### Characterizations

The XRD patterns were recorded by X-ray diffractometer (XRD, PANalytical B.V., Netherlands) with Cu Kα radiation. The scanning electron microscopy (SEM, FEI-Verios G4) was carried out at 15 kV to scrutinize the morphologies of the electrocatalysts. The TEM and HRTEM images were acquired on FEI Talos F200X at an acceleration voltage of 200 kV. The X-ray photoelectron spectroscopy (XPS, Kratos-Axis Supra) was performed to determine the chemical composition of the electrocatalysts. A Nicolet-is50 spectrometer was used to measure the ATR-FTIR with an MCT-A detector. The temperature of detector was controlled using liquid nitrogen. The Cu NAs was pressed onto the Si prism by a glassy carbon electrode to serve as the working electrode. Hg/HgO electrode and platinum electrode were used as the reference and counter electrodes, respectively. 1 M KOH aqueous solution containing 0.5 M MBY was the electrolyte. Data acquisition was performed from 4000 to 650 cm$^{-1}$ with a 4 cm$^{-1}$ nominal resolution and 32 scans for each spectrum. The electrode settings and reaction conditions were kept the same as those in the electrocatalytic tests.

### Theoretical calculations

All the density functional theory (DFT) calculations were carried out by using the Vienna Ab Initio Simulation Package (VASP)[36,37]. The electron

ion interaction was described with the projector augmented wave (PAW) method[38,39], whereas the electron exchange and correlation energy was solved under the generalized gradient approximation by using the revised Perdew-Burke-Ernzerhof (RPBE) exchange-correlation functional[40]. To acquire accurate energies with errors less than 1 meV per atom, an energy cut-off of 400 eV and a second-order Methfessel-Paxton electron smearing with $\sigma = 0.2$ eV were applied. The convergences criteria of optimizations for energy and force were set to be $10^{-5}$ eV and 0.02 eV/Å, respectively. Cu and Pd featuring face-centred cubic (FCC) structure were used to calculate the reaction mechanism of MBY semihydrogenation, and their optimized lattice parameters $(a = b = c)$ were 3.679 Å (Cu) and 3.972 Å (Pd), respectively. A p($3 \times 3$)−4L (100) surface of these metals was used to calculate the reaction mechanism of MBY semihydrogenation. The vertical separation between periodic slabs was set to be 12 Å to avoid obvious interactions and dipole corrections were applied. The DFT computed energies were corrected into free energies with the following equation:

$$\Delta G = \Delta E + \Delta ZPE + \Delta H \text{-} T\Delta S \qquad (2)$$

Where the zero-point energies (ZPE) of adsorbates were from the calculated vibrational frequencies within the harmonic approximation. The enthalpy and entropic contributions were calculated within the harmonic approximation for surface species and the ideal gas approximation for gas phase species.

The free energy change of each step that involves a proton-electron transfer is simulated with computational hydrogen electrode (CHE) model as developed by Nørskov group[41,42], which provides an elegant approach of avoiding the explicit treatment of solvated protons. In this method, proton – electron transfer step $(H^+ + e^- \rightarrow 1/2H_2)$ is in equilibrium with $H_2$ at 0 V, all pH values and 1 atm pressure. Therefore, at zero voltage, $p = 1$ bar, T = 298.15 K, the Gibbs free energy change ($\Delta G_0$) of a reaction (i.e., $R^* + H^+ + e^- \leftrightarrow RH^*$) was described as the free energy of the reaction $R^* + 1/2H_2 \leftrightarrow RH^*$. Then, $\Delta G_0 = \Delta E + \Delta ZPE + \Delta H - T\Delta S$, where $\Delta E = E(\text{product}) - E(\text{reactant})$; $\Delta ZPE$, $\Delta H$ and $\Delta S$ are the differences in zero-point energy, enthalpy and entropy, which are calculated within the harmonic approximation for surface species. The chemical potential of the proton-electron pair could be described as a function of the applied potential, i.e., $\Delta G_U = -eU$, where U is the applied potential. Eventually, the Gibbs free energy changes of a reaction could be calculated with the following formula: $\Delta G = \Delta G_0 + \Delta G_U$.

## Data availability

Additional data are available in the Supplementary Information. Source data are provided with this paper.

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

## Acknowledgements
This work was financially the National Natural Science Foundation of China (22005245), the Natural Science Foundation of Shaanxi Province (2020JQ-141), the Fundamental Research Funds for the Central Universities (3102019JC0005 and G2022KY0606), and the Synergy Innovation Foundation of the University and Enterprise for Graduate Students in Northwestern Polytechnical University (CX2021037 and CX2022074).

## Author contributions
J.Z. designed and supervised the project. J.B. conducted the catalyst syntheses and performance evaluations. J.B., S.C., S.Y., W.M., and Z.L. performed related materials characterizations. J.L. conducted the theoretical calculations. J.B., S.A., Y.W., and Z.L. performed the purification of the MBE product from the electrolyte. J.Z. and J.B. co-wrote the manuscript. All authors discussed the results and commented on the manuscript.

## Competing interests
J.Z., S.C., and J.B. are inventors on patent application CN202111671463.7 submitted by Northwestern Polytechnical University, which covers an electrocatalytic alkynol semi-hydrogenation method for continuous production of alkenols. The other authors declare no competing interests.
