## [Peer Review File · Nature Communications]

REVIEWER COMMENTS

Reviewer #1 (Remarks to the Author):

Zhang J. and colleagues report here on the semihydrogenation of alkynols by electrocatalytic approach on Cu nanoarrays. This work is a particular case of the development of alkyne electrocatalytic hydrogenation, applied to alkynols, an important subclass of alkyne substrates involved in the preparation of fine chemicals (namely in some synthetic pathways to eg retinol). Interestingly, the systems presented here are operated under fully aqueous electrolyte. That feature is likely specific to this class of substrates, owing to the good water-solubility of alkynols and produced alkenols.

The study is clear, and the data generally support the conclusion. Metrics as activity, stability and selectivity are relatively high for the investigated conversions. The novelty of the work is though limited, as this is a particular application of electrocatalytic alkyne semihydrogenation using electrodes based on late TMs (Ni, Cu, Pd), which has been recently extensively reported by these authors and others [Nat Commun 12, 3881 (2021); Nat Catal 4, 565–574 (2021); Nat Commun 12, 6574 (2021); Nat Commun 12, 7072 (2021); ChemSusChem 2022, 15, e202102221; ACS Sust. Chem. Eng. 2019, 7, 11050–11055; ACS Catal. 2021, 11, 9471–9478].

Because this subclass of substrates covers particular synthetic relevance, I yet think that the work is worth for publication in a general journal.

I have only minor specific comments:

- Introduction (l. 37): hydrogen pressures (“1-30 bars”) are commonly used in industrial processes and the associated safety issues dealt with.
- Introduction (l. 39-40): “relatively high reaction temperatures (25-160°C)”. 25°C is not considered as a high reaction temperature.
- How do the systems behave at full conversion? The reported conversions are high (typically >90%), but selectivities should be given at full conversion (>99%). Indeed, the challenge is to keep high selectivity towards alkenols even at full alkyne conversion, as even minor remaining alkyne concentrations would preferentially adsorb (and react) on Cu and hinder overhydrogenation.
- Fig. 3b: the isotopic patterns are not clear and two graphs should be shown separately for the experiments with H₂O and D₂O. Is the peak at m/z 75 due to alcohol H scrambling?
- L. 87-88: “The exponential augment...activity”. This sentence is confusing. Fig. S20 shows that j_{MBE} depends on E_{app}. Do the authors mean that diffusion/transport is not limiting at these potentials?
- L. 93-95: “When the...HER”. This sentence is confusing. What does positively shifted mean for a plot?

- On that last point: the discussion about pH variations deserves more details. Increasing KOH concentration is claimed by the authors to suppress HER, but to increase j_{MBE} at constant applied potential (vs RHE). Why is that so? What is the mechanism and why does increasing KOH increase j_{MBE} at constant driving force? Is alkyne hydrogenation proceeding via surface hydrides or from H⁺/e⁻ transfer to alkynes? These points are also not clearly addressed in the DFT section. Mechanistic considerations should be more thoroughly discussed.
- Can the authors perform isolation of the products from the electrolytic mixture?
- L.49-57-58: "The variations...alkynols" This statement does not agree with data presented in Fig. 4a, as polarization curves for j_{MBE} do depend on flow rates, which points to an effect of mass transfer on alkyne semihydrogenation.

If the authors can address these comments, the work could be considered for publication.

Reviewer #2 (Remarks to the Author):

The manuscript presents an electrocatalytic semi-hydrogenation approach for the production of alkenols. However, I cannot support its publication in Nature Communications.

1. My major concern is the highlight of this manuscript. The authors claimed that "Theoretical and in situ electrochemical infrared investigations reveal that the semi-hydrogenation kinetics are accelerated by exothermic alkynol adsorption and alkenol desorption on the Cu surfaces". I cannot understand the claim of "kinetics are accelerated". Do the authors mean "reaction rate..."? But I cannot find the measurement of reaction rate. In addition, the authors failed to provide the kinetics parameters. Which step is the rate determination step? If the rate determination step is not given, how can the authors conclude the adsorption or desorption equilibrium show effects on the rate?
2. The authors claimed "high temperature" as one of major disadvantage for thermocatalytic semi-hydrogenation of alkynols. However, in literature, the reaction temperature for state-of-the-art Pd catalysts is usually < 40 °C. In addition, I do think "a replacement for the energy-intensive, high-cost, low-efficiency, and hazardous thermocatalytic alkynol hydrogenation process" is over-claimed.
3. The studied Cu catalyst shows a partial current density of 750 mA cm⁻² and a MBE selectivity of 97%. However, the Faradaic Efficiency of MBE is only 58% at -0.88 V vs. RHE, indicating a low energy efficiency. The FE of H₂ is much higher than the previous Cu electrocatalyst in semi-hydrogenation of acetylene <Nat. Catal. 4, 557-564 (2021)>. Can the authors give some explanations about the strong H₂ evolution on Cu NAs?

4. The authors observed a low concentration MBY Dimers at high current densities (-1.3 A cm^{-2}). And, a low potential was beneficial to suppression H₂ evolution. How to determine the optimal potential for alkynols semi-hydrogenation?

5. The DFT calculations show that the MBY molecule is strongly adsorbed on Cu (111) with adsorption energy of -3.2 eV . Did the authors study the leaching of the Cu catalyst from electrode to the electrolyte solution while conducting stability testing?

6. It is interesting that the FE of MBE of Pd nanoparticles kept almost unchanged with the increased current densities in Supplementary Fig. 10. On contrary, the FE of MBE of Cu NAs obviously decreased. This phenomenon should be discussed.

Reviewer #3 (Remarks to the Author):

In this work Zhang et coworkers report an electrocatalytic route for the semi-hydrogenation of alkynols based on Cu catalysts in batch and flow mode with high current densities.

Electrocatalytic routes for organic synthesis are quickly gaining momentum and this paper is a reflection of this trend. From this viewpoint, the topic is likely to grab the interest of a growing audience and the relatively simple equipment needed minimize entry barriers. From a general perspective, the paper is methodologically correct and most conclusions are directly correlated to experimental observations. My general opinion is therefore positive towards publication. I nonetheless find two conceptual missing aspects that I recommend addressing:

- The first concept is how far would this process be from practical implementation with the presented performance. It is clear from Fig. 1d that the vast difference in market price between MBE and MBY would make this process likely profitable. However, I miss a direct comparison with the thermocatalytic process, since the decision between the two of them will be dictated by which one is more efficient. I thus suggest authors add this comparison, which could maybe be guided by the energy efficiency of the two processes.

- Even though it should not be focus of a first communication, I miss more details pertaining catalyst design, aiming at providing more guidelines for the community. The fact that the presented catalyst is copper nanoarrays with a very particular structure makes one wonder if that is a prerequisite. For example, authors could investigate the influence of the initial copper phase, a very well know effect in CO₂ electroreduction. Tests of the initial Cu foam with no post treatment, of the Cu(OH)₂ foam, and of the CuO reduced under H₂ would also provide fruitful guidelines to follow. I also recommend testing the performance of simpler Cu structures like Cu nanoparticles.

Response to Reviewer 1:

Comments:

Zhang J. and colleagues report here on the semihydrogenation of alkynols by electrocatalytic approach on Cu nanoarrays. This work is a particular case of the development of alkyne electrocatalytic hydrogenation, applied to alkynols, an important subclass of alkyne substrates involved in the preparation of fine chemicals (namely in some synthetic pathways to eg retinol). Interestingly, the systems presented here are operated under fully aqueous electrolyte. That feature is likely specific to this class of substrates, owing to the good water-solubility of alkynols and produced alkenols.

The study is clear, and the data generally support the conclusion. Metrics as activity, stability and selectivity are relatively high for the investigated conversions. The novelty of the work is though limited, as this is a particular application of electrocatalytic alkyne semihydrogenation using electrodes based on late TMs (Ni, Cu, Pd), which has been recently extensively reported by these authors and others [Nat Commun 12, 3881 (2021); Nat Catal 4, 565–574 (2021); Nat Commun 12, 6574 (2021); Nat Commun 12, 7072 (2021); ChemSusChem 2022, 15, e202102221; ACS Sust. Chem. Eng. 2019, 7, 11050–11055; ACS Catal. 2021, 11, 9471–9478].

Because this subclass of substrates covers particular synthetic relevance, I yet think that the work is worth for publication in a general journal. I have only minor specific comments:

Response:

We greatly appreciate the Reviewer's positive comments.

Question 1:

Introduction (l. 37): hydrogen pressures (“1-30 bars”) are commonly used in industrial processes and the associated safety issues dealt with.

Response:

We thank the Reviewer for above comment. As suggested by the Reviewer, hydrogen pressures are commonly used in industrial processes and the associated safety issues

are dealt with. However, the treatment for hydrogen safety during utilization and transportation not only increases the cost but also has the potential risk of explosive hydrogen leak.

Accordingly, we have revised “pressurized hydrogen gas (1–30 bar) must be used as the hydrogen source, which leads to serious safety issues” to “pressurized hydrogen gas (1–30 bars) must be used as the hydrogen source, which leads to potential safety issues” in the revised manuscript (page 2).

Question 2:

Introduction (l. 39-40): “relatively high reaction temperatures (25–160°C)”. 25°C is not considered as a high reaction temperature.

Response:

According to the Reviewer’s valuable suggestion, we have revised “relatively high reaction temperatures (25–160 °C)” to “elevated reaction temperatures (<160 °C)” in revised manuscript (page 2).

Question 3:

How do the systems behave at full conversion? The reported conversions are high (typically >90%), but selectivities should be given at full conversion (>99%). Indeed, the challenge is to keep high selectivity towards alkenols even at full alkyne conversion, as even minor remaining alkyne concentrations would preferentially adsorb (and react) on Cu and hinder overhydrogenation.

Response:

Thanks for the Reviewer’s valuable comment. As suggested by the Reviewer, we further evaluated the alkenol selectivity at the full conversion of alkynol. As shown in Fig. R1, the MBE selectivity was >91% at full MBY conversion (>99%).

The related results have been included in the revised manuscript (page 6) and supporting information (Supplementary Fig. 11).

Fig. R1 Time-dependent MBY conversion and MBE selectivity of Cu NAs at -1.3 A cm^{-2} .

Question 4:

Fig. 3b: the isotopic patterns are not clear and two graphs should be shown separately for the experiments with H_2O and D_2O . Is the peak at m/z 75 due to alcohol H scrambling?

Response:

We appreciate the Reviewer's valuable suggestions. Accordingly, Fig. 3b has been separated based on the experiments with H_2O and D_2O (Fig. R2).

The peak at m/z 75 is due to alcohol H/D scrambling. As shown in Fig. R3, compared with MBY in H_2O , a new m/z peak appeared at 71 in D_2O , which originated from H/D scrambling between MBY and D_2O . Thus, after semi-deuteration of MBY, the produced MBE also showed a peak at m/z 75.

Fig. R2 Mass fragmentation patterns of deuterated MBE (C₅H₈D₂O) and MBE (C₅H₁₀O).

Fig. R3 Mass fragmentation patterns of MBY in 1 M KOH aqueous solution, MBY in 1 M KOH D₂O solution, and deuterated MBE (C₅H₈D₂O) in 1 M KOH D₂O solution.

Question 5:

L. 87-88: “The exponential augment...activity”. This sentence is confusing. Fig. S20 shows that j_{MBE} depends on E_{app} . Do the authors mean that diffusion/transport is not limiting at these potentials?

Response:

We are sorry for above spelling mistake. The “independent of” has been corrected to “dependent on”. This result indicates that the influence of MBY mass transport on electrocatalytic activity is minimal [Nat. Commun. 12, 1949 (2021)].

The related correction has been included in the revised manuscript (page 10).

Question 6:

L. 93-95: “When the...HER”. This sentence is confusing. What does positively shifted mean for a plot?

Response:

We are sorry for above unclear description. As suggested by the Reviewer, “the j_{MBE} plot positively shifted” has been corrected to “the j_{MBE} plot shifted toward high potentials”. Specifically, for achieving the same partial current density of MBE, the applied potential is obviously decreased along with increased pH values.

The related correction has been included in the revised manuscript (page 11).

Question 7:

On that last point: the discussion about pH variations deserves more details. Increasing KOH concentration is claimed by the authors to suppress HER, but to increase j_{MBE} at constant applied potential (vs RHE). Why is that so? What is the mechanism and why does increasing KOH increase j_{MBE} at constant driving force? Is alkyne hydrogenation proceeding via surface hydrides or from H^+/e^- transfer to alkynes? These points are also not clearly addressed in the DFT section. Mechanistic considerations should be more thoroughly discussed.

Response:

We thank the Reviewer for above constructive comments. The improvement of j_{MBE} at rising pH values is mainly due to two reasons. First, the electrolyte conductivity increased at higher KOH concentration, which boosted total current density by lowering the resistance of thin electrolyte layer near the catalysts and the bulk electrolyte [Adv. Mater. 30, 1803111 (2018)]. Moreover, higher KOH concentration also suppressed the HER and generated larger j_{MBE} . Second, higher electrolyte pH promoted the water dissociation (Volmer step) [Nat. Commun. 10, 4876 (2019)], which increased the formation and coverage of surface H^* and thus improved j_{MBE} at constant driving force.

As suggested by the Reviewer, we further probe the mechanism of electrochemical MBY hydrogenation. As previously reported, electrocatalytic water dissociation generated chemisorbed hydrogen on the cathode surface via the Volmer process in alkaline solution [Angew. Chem. Int. Ed. 59, 21170-21175 (2020)]. Accordingly, we performed cyclic voltammograms (CV) for investigating the effects of the adsorbed H* on MBY hydrogenation in 1 M KOH solution. As revealed in Fig. R4a, the CV curve in pure 1 M KOH solution showed an anodic peak at about 0.23 V vs. RHE, which was attributed to H* desorption [Angew. Chem. Int. Ed. 61, e202209849 (2022); Angew. Chem. Int. Ed. 59, 21170-21175 (2020)]. By contrast, in 1 M KOH solution containing MBY, the intensity of H* desorption peaks remarkably decreased along with increased MBY concentration, indicating the consumption of surface H* species during MBY hydrogenation. Significantly, the H* desorption peak well recovered in fresh KOH electrolyte (Fig. R4b). Furthermore, we assessed the kinetic of HER and MBY hydrogenation according to the Tafel plots (Fig. R4c). The Tafel slopes of Cu NAs were 114 mV dec⁻¹ for HER and 118 mV dec⁻¹ for MBY hydrogenation, indicating that the Volmer step was the rate-determining step. Therefore, for MBY hydrogenation on Cu NAs, active H* was first generated via the Volmer step. To further verify the MBY hydrogenation mechanism on Cu NAs, we evaluated the reaction rates at high MBY concentrations (0.5 M to 1.5 M). As shown in Fig. R4d, a negative reaction order (-0.46) for MBY was achieved, meaning that the reaction rate declined along with increased MBY concentration. This was attributed to the competitive adsorption of MBY and H₂O molecules on Cu NAs. Therefore, the electrocatalytic MBY hydrogenation should follow the Langmuir–Hinshelwood (L–H) mechanism.

The related results have been included in the revised manuscript (page 11) and supporting information (Supplementary Fig. 23).

Fig. R4. **a**, CV curves of Cu NAs in 1 M KOH solution containing MBY at different concentrations. **b**, CV curves of Cu NAs in KOH electrolyte before and after CV scans in KOH electrolytes containing MBY. **c**, Tafel plots of Cu NAs in KOH electrolyte with or without MBY. **d**, The electrocatalytic productivity of MBE over Cu NAs as a function of MBY concentration in 1 M KOH electrolyte. Reaction conditions: -0.9 V vs. RHE; the MBY conversion of $<20\%$.

Question 8:

Can the authors perform isolation of the products from the electrolytic mixture?

Response:

We thank the Reviewer for valuable suggestion. Accordingly, we performed the separation of MBE from the electrolyte (MBY conversion: 91%; MBE selectivity: 93%). The liquid products were first extracted from the 1 M KOH electrolyte using ether, and was then purified through distillation. The samples were quantified by using gas chromatography (Fuli GC9790Plus) equipped with a flame ionization detector (FID)

(Fig. R5a). The mass fraction of MBE was 81% after the distillation (Fig. R5b), which was consistent with yield of MBE (84%).

The related result has been included in the revised manuscript (page 16) and supporting information (Supplementary Fig. 28)

Fig. R5. **a**, Gas chromatography curves of the samples before and after distillation. **b**, Photograph of the distilled MBE.

Question 9:

L.49-57-58: “The variations...alkynols” This statement does not agree with data presented in Fig. 4a, as polarization curves for j_{MBE} do depend on flow rates, which points to an effect of mass transfer on alkyne semihydrogenation?

Response:

We greatly appreciate the Reviewer’s valuable comment. We are sorry for unclear illustration. First, in Fig. S26, the polarization plots are recorded at different flow rates of 1 M KOH containing MBY, where the MBY is enough sufficient for electrocatalytic semihydrogenation. Accordingly, the variations in polarization curves at different flow rates was negligible, implying that mass transfer is not the limiting factor for the electrocatalytic semi-hydrogenation of alkynols. Second, as shown in Fig. 4a, the large applied voltages and related partial MBE current densities are plotted based on single-pass electrocatalytic process, where the MBY feed is input and the target MBE is output with a high MBY conversion. During such a single-pass process, the MBY content is not sufficient for electrocatalytic semihydrogenation. Thereby, at different flow rates,

the j_{MBE} reached a plateau along with increased voltages because the MBY had been mostly used up. Therefore, the large j_{MBE} was achieved at high flow rates because more MBY was provided, which did not reflect the mass transport.

Response to Reviewer 2:

Comments:

The manuscript presents an electrocatalytic semi-hydrogenation approach for the production of alkenols. However, I cannot support its publication in Nature Communications.

Response:

We appreciate the Reviewer's comments.

Question 1:

My major concern is the highlight of this manuscript. The authors claimed that "Theoretical and in situ electrochemical infrared investigations reveal that the semi-hydrogenation kinetics are accelerated by exothermic alkynol adsorption and alkenol desorption on the Cu surfaces". I cannot understand the claim of "kinetics are accelerated". Do the authors mean "reaction rate..."? But I cannot find the measurement of reaction rate. In addition, the authors failed to provide the kinetics parameters. Which step is the rate determination step? If the rate determination step is not given, how can the authors conclude the adsorption or desorption equilibrium show effects on the rate?

Response:

We greatly appreciate the Reviewer's constructive comments. We are sorry for unclear description. According to the Reviewer's suggestion, "Theoretical and in situ electrochemical infrared investigations reveal that the semi-hydrogenation kinetics are accelerated by exothermic alkynol adsorption and alkenol desorption on the Cu surfaces" has been revised to "Theoretical and in situ electrochemical infrared investigations reveal that the semi-hydrogenation performance is enhanced by exothermic alkynol

adsorption and alkenol desorption on the Cu surfaces” in the revised manuscript (page 2). Similar illustrations have also been given in recent literatures [Science 320, 1320-1322 (2008); Nat. Catal. 4, 565-574 (2021)].

Based on the theoretical simulations (Fig. 3a), the Gibbs free energy of MBY adsorption on Cu surface was higher than that on Pd. However, the MBY adsorption on Cu surface ($\Delta G = -0.31$ eV) was considerably exothermic, which facilitated the adsorption and activation of MBY molecules. Furthermore, the desorption of MBE from Cu surface ($\Delta G = -0.69$ eV) was energetically favourable, but that from Pd surface was endothermic ($\Delta G = 0.10$ eV). Therefore, the semi-hydrogenation of MBY is promoted on Cu surfaces.

As suggested by the Reviewer, we further investigated the mechanism of electrochemical MBY hydrogenation. As previously reported, electrocatalytic water dissociation generated chemisorbed hydrogen on the cathode surface via the Volmer process in alkaline solution [Angew. Chem. Int. Ed. 59, 21170-21175 (2020)]. Accordingly, we performed cyclic voltammograms (CV) for investigating the effects of the adsorbed H* on MBY hydrogenation in 1 M KOH solution. As revealed in Fig. R4a, the CV curve in pure 1 M KOH solution showed an anodic peak at about 0.23 V vs. RHE, which was attributed to H* desorption [Angew. Chem. Int. Ed. 61, e202209849 (2022); Angew. Chem. Int. Ed. 59, 21170-21175 (2020)]. By contrast, in 1 M KOH solution containing MBY, the intensity of H* desorption peaks remarkably decreased along with increased MBY concentration, indicating the consumption of surface H* species during MBY hydrogenation. Notably, the H* desorption peak well recovered in fresh KOH electrolyte (Fig. R4b). Furthermore, we assessed the kinetic of HER and MBY hydrogenation according to the Tafel plots (Fig. R4c). The Tafel slopes of Cu NAs were 114 mV dec⁻¹ for HER and 118 mV dec⁻¹ for MBY hydrogenation, indicating that the Volmer step was the rate-determining step. Therefore, for MBY hydrogenation on Cu NAs, active H* was first generated via the Volmer step. To further verify the MBY hydrogenation mechanism on Cu NAs, we evaluated the reaction rates at high MBY concentrations (0.5 M to 1.5 M). As shown in Fig. R4d, a negative reaction order (-0.46) for MBY was achieved, meaning that the reaction rate declined along with

increased the MBY concentration. This was attributed to the competitive adsorption of MBY and H₂O molecules on Cu NAs. Therefore, the electrocatalytic MBY hydrogenation should follow the Langmuir–Hinshelwood (L–H) mechanism.

The related results have been included in the revised manuscript (pages 2 and 11) and supporting information (Supplementary Fig. 23).

Fig. R6. **a**, CV curves of Cu NAs in 1 M KOH solution containing different concentrations of MBY. **b**, CV curves of Cu NAs in KOH electrolyte before and after CV scans in KOH electrolytes containing MBY. **c**, Tafel plots of Cu NAs in KOH electrolyte with or without MBY. **d**, The electrocatalytic productivity of MBE over Cu NAs as a function of MBY concentration in 1 M KOH electrolyte. Reaction conditions: -0.9 V vs. RHE; the MBY conversion of $<20\%$.

Question 2:

The authors claimed “high temperature” as one of major disadvantage for thermocatalytic semi-hydrogenation of alkynols. However, in literature, the reaction temperature for state-of-the-art Pd catalysts is usually < 40 °C. In addition, I do think “a replacement for the energy-intensive, high-cost, low-efficiency, and hazardous thermocatalytic alkynol hydrogenation process” is over-claimed.

Response:

We thank the Reviewer for above comments. We are sorry for the unclear demonstration. According to the suggestion of the Reviewer, “relatively high reaction temperatures (25–160 °C)” has been revised to “elevated reaction temperatures (< 160 °C)”. As shown in Table. R1, the reported reaction temperatures of thermocatalytic semi-hydrogenation are different for different alkynols and catalysts. Although several literatures reported the thermocatalytic semi-hydrogenation at room temperature, but they required much longer reaction time and larger amounts of catalysts. Similarly, some literatures also reported the thermocatalytic semi-hydrogenation under 1 bar H_2 , but H_2 balloons and bubbling are essential for feeding excessive hydrogen source. By contrast, the electrocatalytic alkynol hydrogenation process was operated at room temperature on Cu electrocatalysts with water as hydrogen source. In addition, the energy consumption of the thermocatalytic MBE production was estimated to be $7.73 \text{ kW h}^{-1} \text{ kg}^{-1}$ (950 kgce/t) according to similar energy consumption of butynediol hydrogenation. However, for continuous 8-hour electrochemical production of MBE at a cell current of 1.8 A ($E_{\text{cell}} \approx 2.89 \text{ V}$), the energy consumption was only $3.39 \text{ kW h}^{-1} \text{ kg}^{-1}$. Meanwhile, the thermocatalytic process was usually carried out in pressurized (2–30 bars) liquid phase reactors, which increased the number of processing steps and thus limited the production efficiency [ACS Appl. Mater. Interfaces 12, 28158-28168 (2020)]. Here, we perform the electrocatalytic hydrogenation of MBY in a home-made flow electrolyser with a large electrode area of 25 cm^2 , shifting the typical batch process to continuous operation. As a result, we achieved a single-pass MBY conversion of $\sim 91\%$ and MBE selectivity of $\sim 90\%$ at an electrolyte (0.5 M MBY in 1 M KOH) flow rate of 134 mL h^{-1} . The production rate of MBE reached $169 \text{ g g}_{\text{Cu}}^{-1} \text{ h}^{-1}$, which was considerably higher than

the rates of reported thermocatalysts. Therefore, in comparison with utilization of explosive hydrogen, elevated temperature, and noble-metal Pd-based thermocatalysts in thermocatalytic alkyne hydrogenation, electrocatalytic alkyne hydrogenation is a relatively energy-saving, low-cost, high-efficiency, and safety process. Based on the suggestion of the Reviewer, “a replacement for the energy-intensive, high-cost, low-efficiency, and hazardous thermocatalytic alkyne hydrogenation process” has been revised to “a replacement for relatively energy-intensive, high-cost, low-efficiency, and hazardous thermocatalytic alkyne hydrogenation process”.

The related results have been included in the revised manuscript (pages 2 and 16).

Table. R1. The comparison of alkynes semi-hydrogenation performance between electrocatalytic and thermocatalytic processes.

Catalysts	Alkynes	H ₂ pressure (bar)	Temperature (°C)	Time (h)	Reference
Cu nanoarrays (3.3 mg)	2-Methyl-3-butyne-2-ol (10 mmol)	0	25	1	this work (three electrode) (1.3 A cm ⁻²)
Pt/SiC (50 mg)	2-Butyne-1,4-diol (12 mmol)	10	100	10	Catal. Sci. Technol. 10 , 327-331 (2020)
Pd/ZnO (30 mg)	2-Methyl-3-butyne-2-ol (40 mmol)	1 (bubbling)	70	3	Catal. Today 273 , 205-212 (2016)
Pt@ZIF-8 (50 mg)	2-Butyne-1,4-diol (6.6 mmol)	30	120	6	Chin. J. Catal. 37 , 1555-1561 (2016)
Pd(B,C) (8 mg)	2-Methyl-3-butyne-2-ol (1.25 mmol)	1 (balloon)	25	0.8	Nat. Commun. 13 , 2754 (2022)
Pd/ZnO (15 mg)	2-Methyl-3-butyne-2-ol (20 mmol)	5	35	3	J. Catal 251 , 213-222 (2007)
PdZn/CN@ZnO (40 mg)	2-Methyl-3-butyne-2-ol (5 mmol)	5	35	1.6	J. Catal 350 , 13-20 (2017)
Pd-In/In ₂ O ₃ (20 mg)	2-Methyl-3-butyne-2-ol (1.25 mmol)	1 (balloon)	80	1.2	Green Chem. 21 , 4143-4151 (2019)
Pd/TiO ₂ -V ₂ O ₅ @C (20 mg)	2-Methyl-3-butyne-2-ol (8 mmol)	2	25	0.5	ACS Catal. 9 , 10656-10667 (2019)
Pd-DETA (5 mg)	2-Methyl-3-butyne-2-ol (1 mmol)	1 (balloon)	35	0.5	ACS Appl. Mater. Interfaces 13 , 31775-31784 (2021)

Pd nanocube	2-Methyl-3-butyn-2-ol	3	60	-	J. Am. Chem. Soc. 133 , 12787-12794 (2011)
PdCu/ZnO	Dehydroisophytol	4	80	-	Appl. Catal. A: Gen. 478 , 186-193 (2014)

Question 3:

The studied Cu catalyst shows a partial current density of 750 mA cm^{-2} and a MBE selectivity of 97%. However, the Faradaic Efficiency of MBE is only 58% at -0.88 V vs. RHE, indicating a low energy efficiency. The FE of H_2 is much higher than the previous Cu electrocatalyst in semi-hydrogenation of acetylene (<Nat. Catal. 4, 557-564 (2021)>). Can the authors give some explanations about the strong H_2 evolution on Cu NAs?

Response:

We thank the Reviewer for above comment. For the increased H_2 evolution at a large MBE current density of 750 mA cm^{-2} (-0.88 V vs. RHE), there are two major reasons:

1) At large current density, the competition of HER increased. The Cu dendrites (prepared by using the same method in Nat. Catal. 4, 557-564 (2021)) showed stronger H_2 evolution (e.g., $\text{FE} = 43\%$ at -1.3 A cm^{-2}) than Cu NAs ($\text{FE} = 39\%$) for the MBY hydrogenation (Fig. R7). The acetylene hydrogenation occurs at gas-solid-liquid tri-phase interface with pure acetylene. By contrast, the MBY hydrogenation occurs at solid-liquid bi-phase interface, where MBY concentration is greatly lower than pure acetylene.

2) The current density for MBY hydrogenation is 1.3 A cm^{-2} at -0.88 V vs. RHE, which was considerably larger than $\sim 0.2 \text{ A cm}^{-2}$ for reported acetylene hydrogenation. At greatly increased current density, the HER was also enhanced due to the rising H^* coverage on Cu NAs.

Fig. R7. The H₂ FEs for Cu NAs and Cu dendrites at different current densities.

Question 4:

The authors observed a low concentration MBY Dimers at high current densities (-1.3 A cm^{-2}). And, a low potential was beneficial to suppression H₂ evolution. How to determine the optimal potential for alkynols semi-hydrogenation?

Response:

We thank the Reviewer for above comment. The mass spectrometry detected a particularly low concentration MBY dimers at high current densities (-1.3 A cm^{-2}) (Fig. R8). The selectivity of MBY dimers was measured to be only $\sim 0.8\%$, which was far lower than 97% for MBE. In addition, it's easy to separate MBY dimers from MBE by rectification because the boiling point of the MBY dimers ($286 \text{ }^\circ\text{C}$) is much higher than that of MBE ($98 \text{ }^\circ\text{C}$). As a gas phase by-product, the generated H₂ can be directly separated from the electrolyte. Thus, we comprehensively considered the MBY conversion, MBE selectivity and partial current density of MBE as the important parameters for alkynol semi-hydrogenation. Accordingly, under current operation conditions, -0.88 V vs. RHE was the optimal potential for MBY semi-hydrogenation, where the MBE current density reached 750 mA cm^{-2} with a MBE production rate of $357 \text{ g g}_{\text{Cu}}^{-1}\text{h}^{-1}$ (Fig. R9).

Fig. R8. Total ion chromatogram (TIC) versus retention time of the GC-MS analysis.

Fig. R9. The potential-dependent partial current densities of MBE.

Question 5:

The DFT calculations show that the MBY molecule is strongly adsorbed on Cu (111) with adsorption energy of -3.2 eV. Did the authors study the leaching of the Cu catalyst from electrode to the electrolyte solution while conducting stability testing?

Response:

Based on the Reviewer's suggestion, we measured the concentration of Cu in the electrolyte after the long-term stability test by using inductively coupled plasma atomic emission spectrometry (ICP-AES). The concentration of Cu was measured to be 0.241 $\mu\text{g mL}^{-1}$ after the continuous 20-hours hydrogenation of MBY.

Question 6:

It is interesting that the FE of MBE of Pd nanoparticles kept almost unchanged with the increased current densities in Supplementary Fig. 10. On contrary, the FE of MBE of Cu NAs obviously decreased. This phenomenon should be discussed?

Response:

Thanks for the Reviewer's valuable comment. As shown in Supplementary Fig. 10, when the applied potential changed from -0.3 to -1.3 V, the FE_{MBE} of Pd remained about 42%. By contrast, the MBE FE of Cu NAs gradually decreased from 78% at -0.3 V to 58% at -1.3 V, which were much higher than those for Pd. This phenomenon was attributed to stronger adsorption of MBY molecules on Pd than Cu. As indicated in Fig. 3c, the free energy of adsorbed MBY ($*C_5H_8O$) on Pd(111) is -1.50 eV, which is much lower than -0.31 eV on Cu(111). Therefore, As the applied current density increased, water adsorption and formed H^* strongly compete with the relatively weak MBY adsorption on Cu in comparison with strong MBY adsorption on Pd. As a result, the FE_{MBE} of Pd nanoparticles kept almost unchanged with increased current densities. However, the partial current density and FE_{MBE} of Pd are substantially lower than those for Cu.

Supplementary Figure 10. The electrocatalytic MBY semi-hydrogenation performance of Cu NAs and Pd nanoparticles. a, partial current densities and b, Faradaic efficiency of MBE versus the applied current density.

Response to Reviewer 3:

Comments:

In this work Zhang et coworkers report an electrocatalytic route for the semi-hydrogenation of alkynols based on Cu catalysts in batch and flow mode with high current densities.

Electrocatalytic routes for organic synthesis are quickly gaining momentum and this paper is a reflection of this trend. From this viewpoint, the topic is likely to grab the interest of a growing audience and the relatively simple equipment needed minimize entry barriers. From a general perspective, the paper is methodologically correct and most conclusions are directly correlated to experimental observations. My general opinion is therefore positive towards publication. I nonetheless find two conceptual missing aspects that I recommend addressing:

Response:

We greatly appreciate the Reviewer's positive comments.

Question 1:

The first concept is how far would this process be from practical implementation with the presented performance. It is clear from Fig. 1d that the vast difference in market price between MBE and MBY would make this process likely profitable. However, I miss a direct comparison with the thermocatalytic process, since the decision between the two of them will be dictated by which one is more efficient. I thus suggest authors add this comparison, which could maybe be guided by the energy efficiency of the two processes.

Response:

Thanks for the Reviewer's valuable comment and suggestion. As shown in Fig. R10, we further compare the energy consumption for producing 1 kg MBE using electrocatalytic and thermocatalytic processes. The energy consumption of thermocatalytic MBE production was estimated to be $7.73 \text{ kW h}^{-1} \text{ kg}^{-1}$ (950 kgce/t) according to similar energy consumption of butynediol hydrogenation. The energy

consumption of electrocatalytic MBE production mainly changed from 2 to 6 kW h⁻¹ kg⁻¹ under different conditions, which was substantially lower than 7.73 kW h⁻¹ kg⁻¹ for the thermocatalytic process. Especially, for continuous 8-hour electrochemical production of MBE at a cell current of 1.8 A ($E_{\text{cell}} \approx 2.89$ V), the energy consumption was only 3.39 kW h⁻¹ kg⁻¹, which was about 44% of those for the thermocatalytic process. Therefore, the electrocatalytic approach is more energy-efficient than the thermocatalytic process.

The related results have been included in the revised supporting information (Supplementary Figure 30, page 31).

Fig. R10. The energy consumption of 1 kg MBE at different cell voltages and electrolyte flow rates.

Question 2:

Even though it should not be focus of a first communication, I miss more details pertaining catalyst design, aiming at providing more guidelines for the community. The fact that the presented catalyst is copper nanoarrays with a very particular structure makes one wonder if that is a prerequisite. For example, authors could investigate the influence of the initial copper phase, a very well know effect in CO₂ electroreduction. Tests of the initial Cu foam with no post treatment, of the Cu(OH)₂ foam, and of the CuO reduced under H₂ would also provide fruitful guidelines to follow. I also recommend testing the performance of simpler Cu structures like Cu nanoparticles.

Response:

Thanks for the Reviewer's valuable suggestions. Accordingly, we further measured the electrocatalytic MBY hydrogenation performance of above-mentioned Cu catalysts. The j_{partial} of MBE increased gradually over all of the samples with improved applied current density (Fig. R11a). At an ultrahigh current density of 1.3 A cm^{-2} , Cu NAs showed the highest j_{MBE} of 750 mA cm^{-2} , which was much higher than 370 mA cm^{-2} for Cu foam. The j_{MBE} showed tiny difference between $\text{Cu}(\text{OH})_2$ NAs, CuO NAs-H_2 and Cu NPs, which was about 620 mA cm^{-2} . This reveals the influence of specific surface area of Cu catalysts on partial MBE current density. In addition, no large difference was observed for the selectivity of MBE on different Cu-based electrocatalysts (Fig. R11b).

Fig. R11. **a**, The partial current density and **b**, specific selectivity of MBE on different Cu catalysts in the 1 M KOH solution containing 0.5 M MBY.

REVIEWER COMMENTS

Reviewer #1 (Remarks to the Author):

In their revised manuscript, Zhang J. and colleagues have addressed satisfactorily most of the comments from the reviewers.

I have only specific points remaining:

- The selectivity at full conversion is good but not excellent. Especially, the purity of the isolated compound (Fig. R5) provided in the revised version is quite low for this kind of compounds. This point questions the applicability of the approach. Can alkenols be obtained in high purity (>99%) by this approach?
- The explanation for the pH dependence of j_{MBE} (KOH concentration; Fig. S22) is not fully satisfactory. It would be expected that the solution resistance is compensated for in such setup. To check whether the solution resistance has really an impact here, an experiment at increasing ionic strength but identical pH may be useful.
- Concerning transport limitation: most likely there are two regimes. Transport is not limiting at low voltages (Fig. S26), but is at high voltages (Fig. 4a).

Once the comments addressed, the work can be accepted.

Reviewer #2 (Remarks to the Author):

The authors have added some new experimental data and discussion in this revised manuscript to address my comments. However, I still have concerns for the highlight of the revised manuscript.

1. The authors claimed that the H₂O dissociation is the RDS. But the increase of the RDS rate would also lead to the improvement of side reaction of HER. But the authors failed to mention this effect.

2. The authors claimed that the reaction follows L-H mechanism, water adsorption and formed H^* strongly compete with the relatively weak MBY adsorption on Cu, and the H_2O dissociation step is RDS. Why is the reaction order for MBY about -0.5? Have the authors considered any other possibilities?

Reviewer #3 (Remarks to the Author):

Authors addressed my first concern properly. I must highlight, however, a my discomfort with the reply to my second one.

Authors performed the screening of Cu materials I suggested, revealing the very mild dependence on the structure of the Cu catalysts. Selectivity was unaffected by it, and activity may also behave similarly if it is normalized to the electrochemical surface area of the electrodes. If that is the case, the use of nanoarrays would be of very limited fundamental relevance, if any, apart from increasing the exposed area per cm^2 .

I think authors did not add any action on the manuscript on this regard. Authors should discuss this aspect and if the reaction is totally or largely insensitive to the Cu structure, claim it openly. I find this information of high relevance for future efforts on catalyst and electrode design for this reaction.

Response to Reviewer 1:

Comments:

In their revised manuscript, Zhang J. and colleagues have addressed satisfactorily most of the comments from the reviewers.

I have only specific points remaining:

Response:

We greatly appreciate the Reviewer's positive comments.

Question 1:

The selectivity at full conversion is good but not excellent. Especially, the purity of the isolated compound (Fig. R5) provided in the revised version is quite low for this kind of compounds. This point questions the applicability of the approach. Can alkenols be obtained in high purity (>99%) by this approach?

Response:

We thank the Reviewer for above comment. In our previous manuscript, the liquid products were first extracted from the 1 M KOH electrolyte using ether, and were then purified through simple vacuum distillation. After distillation, there were residual ether, 2-methyl-3-butyne-2-ol (MBY) and 2-methyl-3-butan-2-ol (MBA). Based on the valuable suggestion of the Reviewer, we further discussed with several senior engineers in Hualu Engineering and Technology Co., Ltd. The preparative chromatography was subsequently utilized for purifying the distilled product. The purified product was quantified by using gas chromatography (Fuli GC9790Plus) equipped with a flame ionization detector (FID). As shown in Fig. R1, the mass fraction of MBE reached as high as 98% (Fig. R1), which was same with the commercial MBE (98%) e.g., Sigma Aldrich (98%), Shanghai Macklin Biochemical Co., Ltd (98%).

The related results have been included in the revised manuscript (page 16) and supporting information (Supplementary Figure 31).

Fig. R1 Gas chromatography curves of the purchased standard MBE (Shanghai Macklin Biochemical Co., Ltd) and the products before and after purification using preparative chromatography.

Question 2:

The explanation for the pH dependence of j_{MBE} (KOH concentration; Fig. S22) is not fully satisfactory. It would be expected that the solution resistance is compensated for in such setup. To check whether the solution resistance has really an impact here, an experiment at increasing ionic strength but identical pH may be useful.

Response:

Thanks for the Reviewer's constructive comment. As suggested, the solution resistance effect was further investigated by changing the electrical conductivity at identical pH value. Thus, different amount of KCl was added into 1 M KOH electrolyte. As shown in Fig. R2, the pH value of the electrolytes was well kept at ~ 13.7 , but the electrical conductivity increased from 180 mS cm^{-1} for 1 M KOH to 225 mS cm^{-1} for 1 M KOH + 0.5 M KCl and 270 mS cm^{-1} for 1 M KOH + 1 M KCl. Then, electrocatalytic MBY hydrogenation was performed in these electrolytes. As revealed in Fig. R3, the variations of MBY hydrogenation performance (total current density: $\sim 830 \text{ mA cm}^{-2}$; F_{EMBE} : $\sim 60\%$; j_{MBE} : $\sim 500 \text{ mA cm}^{-2}$) were negligible. These results confirmed that solution resistance had negligible influence on MBY hydrogenation performance. Therefore, the improvement of j_{MBE} at rising pH values is mainly due to the accelerated

water dissociation (Volmer step) [*Nat. Commun.* **10**, 4876 (2019)], which enhanced the formation and coverage of surface H* and thus improved j_{MBE} at constant voltage.

The related results have been included in the revised manuscript (page 11 and Supplementary Figure 25).

Fig. R2 The pH values and electrical conductivity of the KOH electrolyte without and with KCl.

Fig. R3. **a**, Current density profiles for MBY hydrogenation on Cu NAs in different electrolytes. **b**, The FE and j_{partial} of MBE in different electrolytes. Reaction conditions: -0.8 V vs. RHE; 0.5 M MBY in electrolyte.

Question 3:

Concerning transport limitation: most likely there are two regimes. Transport is not limiting at low voltages (Fig. S26), but is at high voltages (Fig. 4a).

Response:

We greatly appreciate the Reviewer's valuable suggestion. As suggested, the transport is not limiting at low voltages because the MBY is sufficient. The transport is limiting at high voltages because the MBY is insufficient. Similarly, in Supplementary Fig. 29, the polarization plots were recorded at different flow rates of 1 M KOH containing sufficient MBY for electrocatalytic semihydrogenation. Accordingly, the variations of polarization curves at different flow rates were negligible, implying that the transport was not the limiting factor for the electrocatalytic semihydrogenation of alkynols. As shown in Fig. 4a, the large applied voltages and related partial MBE current densities are plotted based on single-pass electrocatalytic process, where the MBY feed is input and the target MBE is output at a high MBY conversion. During such a single-pass process, there was a maximum j_{MBE} , which was dependent on the MBY concentration and flow rate of electrolyte. Thereby, at a constant flow rate, the j_{MBE} reached a plateau along with increased voltages because the MBY had been used up. The large j_{MBE} was achieved at high flow rates because more MBY was provided.

According to the Reviewer's suggestion and to avoid misunderstanding, we have deleted the sentence "implying that mass transfer was not the limiting factor of the electrocatalytic semihydrogenation of alkynols" in the revised manuscript (page 14).

Response to Reviewer 2:**Comments:**

The authors have added some new experimental data and discussion in this revised manuscript to address my comments. However, I still have concerns for the highlight of the revised manuscript.

Response:

We appreciate the Reviewer's comments.

Question 1:

The authors claimed that the H₂O dissociation is the RDS. But the increase of the RDS rate would also lead to the improvement of side reaction of HER. But the authors failed to mention this effect.

Response:

We greatly appreciate the Reviewer's constructive comments. In our previous manuscript, we speculated that the improvement of j_{MBE} at rising pH values was mainly due to the promoted water dissociation (Volmer step) [Nat. Commun. **10**, 4876 (2019)], which enhanced the formation and coverage of surface H* and thus improved j_{MBE} at constant voltage. According to the suggestion of the Reviewer, we further evaluated the HER selectivity in different pH values. As shown in Fig. R4b, when the concentration of KOH electrolyte was changed from 0.5 M to 3 M, the FE of H₂ first decreased in 1 M KOH and then increased in 2 M and 3 M KOH solution. Clearly, along with increased pH values, the FE changes of MBE and H₂ was opposite, indicating their strong competition. Therefore, after the formation of surface H*, the competition between MBE hydrogenation and H₂ production occurred. Based on the selectivity, the optimal concentration of KOH was 1 M.

The related results have been included in Supplementary Figure 24.

Fig. R4 The FE of (a) MBE and (b) H₂ in different alkaline electrolyte.

Question 2:

The authors claimed that the reaction follows L-H mechanism, water adsorption and formed H* strongly compete with the relatively weak MBY adsorption on Cu, and the H₂O dissociation step is RDS. Why is the reaction order for MBY about -0.5? Have the authors considered any other possibilities?

Response:

We greatly appreciate the Reviewer's constructive comment and we are sorry for our unclear illustration. As shown in Supplementary Fig. 21, the adsorption energy of MBY on the Cu(111) surface was -3.20 eV, which was much lower than -0.33 eV for H₂O. Thus, the MBY adsorption was substantially stronger than the H₂O adsorption on Cu. The increased MBY concentrations from 0.5 M to 1.5 M extended the MBY coverage on the Cu surfaces, which inherently suppressed H₂O adsorption and subsequent dissociation. As a result, along with increased the MBY concentration, the formation and coverage of surface H* decreased, which led to a negative reaction order of -0.46 for MBY.

Response to Reviewer 3:

Comments:

Authors addressed my first concern properly. I must highlight, however, a my discomfort with the reply to my second one.

Authors performed the screening of Cu materials I suggested, revealing the very mild dependence on the structure of the Cu catalysts. Selectivity was unaffected by it, and activity may also behave similarly if it is normalized to the electrochemical surface area of the electrodes. If that is the case, the use of nanoarrays would be of very limited fundamental relevance, if any, apart from increasing the exposed area per cm².

I think authors did not add any action on the manuscript on this regard. Authors should discuss this aspect and if the reaction is totally or largely insensitive to the Cu structure, claim it openly. I find this information of high relevance for future efforts on catalyst and electrode design for this reaction.

Response:

We sincerely apologize for our previous misunderstanding. We have added the related discussion in the revised manuscript: “In addition, electrocatalytic MBY hydrogenation performance of different Cu-based catalysts was evaluated. Obviously, the MBE selectivity and the electrochemical active surface area (ECSA) normalized j_{MBE} of different Cu-based electrocatalysts were similar. This result reveals that the electrocatalytic MBY hydrogenation performance is largely insensitive to the Cu structure (Supplementary Fig. 14).”.

According to the Reviewer’s valuable suggestion, we further evaluated the j_{MBE} normalized by the electrochemical active surface area (ECSA) of different Cu-based catalysts. Here, double layer capacitance (C_{dl}) was first measured to estimate the ECSA of Cu-based catalysts (Fig. R5a). The Cu NAs showed the highest C_{dl} (24.5 mF cm^{-2}) among various Cu-based catalysts, which agreed well with the j_{partial} of MBE. Obviously, the ECSA normalized j_{MBE} of Cu NAs, CuO NA- H_2 and Cu NPs were similar. Meanwhile, no obvious difference was observed for the MBE selectivity of different Cu-based electrocatalysts. These results suggest that the specific surface area mainly influences partial MBE current density for different Cu-based catalysts (Fig. R5b).

The related results have been included in the revised manuscript (page 7) and supporting information (Supplementary Figure 14).

Fig. R5. Comparison of (a) C_{dl} and (b) ECSA normalized partial MBE current densities for different Cu-based catalysts.

REVIEWERS' COMMENTS

Reviewer #1 (Remarks to the Author):

In this second revision of their manuscript, Zhang J. and colleagues have addressed all the comments I had. In my opinion, the work can be accepted for publication.

Reviewer #2 (Remarks to the Author):

The authors have addressed all my comments in a convincing way. The paper is now well-rounded and I would recommend the manuscript for its publication in Nature Communications.

Reviewer #3 (Remarks to the Author):

Authors have addressed my concerns in their revised version.

Response to Reviewer 1:**Comments:**

In this second revision of their manuscript, Zhang J. and colleagues have addressed all the comments I had. In my opinion, the work can be accepted for publication.

Response:

We greatly appreciate the Reviewer's positive comments.

Response to Reviewer 2:**Comments:**

The authors have addressed all my comments in a convincing way. The paper is now well-rounded and I would recommend the manuscript for its publication in Nature Communications.

Response:

We greatly appreciate the Reviewer's positive comments.

Response to Reviewer 3:**Comments:**

Authors have addressed my concerns in their revised version.

Response:

We greatly appreciate the Reviewer's positive comments.